# Acute Effects of Diaphragmatic Breathing on Trunk and Shoulder Mobility and Pulmonary Function in Healthy Young Adults

**DOI:** 10.3390/jfmk10030325

**Published:** 2025-08-23

**Authors:** Ana Ristovski, Marko Kapeleti, Igor Zlatović, Vladimir Mrdaković

**Affiliations:** Faculty of Sport and Physical Education, University of Belgrade, Blagoja Parovića 156, 11030 Belgrade, Serbia; ana.ristovski@fsfv.bg.ac.rs (A.R.); marko.kapeleti@fsfv.bg.ac.rs (M.K.); igor.zlatovic@fsfv.bg.ac.rs (I.Z.)

**Keywords:** respiration, lung, chest, thoracic spine, spirometry, respiratory health

## Abstract

**Background:** This study investigated whether diaphragmatic breathing intervention could lead to acute improvements in trunk and shoulder mobility and pulmonary function in healthy young adults. **Methods:** Twenty-six physically active males (aged 24.3 ± 2.0 years, body height of 182.9 ± 6.4 cm, and body weight of 82.8 ± 10.4 kg) were randomly assigned to either an experimental or a control group. The experimental group underwent a 22 min diaphragmatic breathing intervention in a lying position. The control group lay passively, breathing naturally. Mobility assessments (chest expansion, thoracic spine rotation, lateral trunk flexion, and shoulder girdle mobility) and pulmonary function tests (forced vital capacity, forced expiratory volume in one second and their ratio) were conducted before and after the intervention. **Results:** Only experimental group showed significant improvements after the intervention (*p* ≤ 0.01) in the chest expansion (+22.2%, ES = 0.62), thoracic spine rotation (+21.7%, ES = 0.76 on the left and +23.3%, ES = 0.84 on the right side), lateral trunk flexion (+11.7%, ES = 0.62 on the left and +15.4%, ES = 1 on the right side), shoulder girdle mobility (+20.2%, ES = 0.44 on the left and +21.5%, ES = 0.38 on the right side), forced vital capacity (+4.7%, ES = 0.39) and reduction (*p* ≤ 0.01) in ratio between forced expiratory volume in one second and forced vital capacity (−4.6%, ES = 0.47). **Conclusion:** The results revealed that a 22 min diaphragmatic breathing intervention could immediately improve trunk and shoulder mobility and pulmonary function, likely due to anatomical relationships and more efficient use of respiratory muscles, especially the diaphragm.

## 1. Introduction

Breathing is a fundamental physiological function essential for supplying oxygen and removing carbon dioxide and hydrogen [1]. In addition to facilitating gas exchange between the organism and the external environment, breathing patterns indirectly influence multiple physiological systems, including the musculoskeletal, cardiovascular, and nervous systems [2,3,4]. Moreover, diaphragmatic breathing can acutely reduce cortisol, improve psychological well-being, and serve as a low-cost, self-regulated stress management strategy [5,6]. Breathing exercises optimize respiratory muscle function, increase respiratory efficiency, and contribute to joint mobility and lung function. Optimal joint mobility is crucial for maintaining overall health and functional capacity. In particular, the mobility of the chest, thoracic and lumbar spine, and shoulder girdle plays a significant role in preventing injury, reducing pain [7,8,9], and maintaining lung function and respiratory health [10].

Breathing relies on the sternum, ribs, thoracic vertebrae, intervertebral disks, bony joints, muscles, and ligaments. The ribs articulate with the thoracic vertebrae at the costovertebral and costotransverse joints, whereas the true ribs connect to the sternum via the sternocostal joints. Thoracic spine movement directly influences shoulder positioning through the scapulohumeral and scapulothoracic joints. During inspiration, rib movement facilitates the expansion of the thoracic cavity in both the lateral and anteroposterior dimensions [11]. The primary inspiratory muscles include the diaphragm, the external intercostal muscles, and the scalene muscles, which work together to expand the thoracic cavity during inhalation. The diaphragm is the most important respiratory muscle, providing 60 to 80 percent of the breathing force through its ability to increase the thoracic volume in three dimensions [12]. Additionally, the elevation of the upper ribs raises the sternum, further increasing the thoracic volume. Simultaneously, the diaphragm contracts and descends toward the abdominal cavity, pulling the lower ribs downward and increasing intra-abdominal pressure. During passive exhalation, the elastic recoil of the cartilaginous structures and muscle relaxation restore the chest to its neutral position while the diaphragm ascends, reducing thoracic volume and facilitating exhalation. In contrast, during forced exhalation, the internal intercostals and abdominal muscles (rectus abdominis, obliques, and transversus abdominis) are actively engaged to reduce thoracic volume and facilitate air expulsion.

Optimal breathing mechanics involve diaphragmatic breathing, which ensures proper activation of the diaphragm and lower abdominal muscles, expansion of the lower chest and efficient use of respiratory muscles, as confirmed by surface and esophageal electromyography recordings [13,14]. This breathing pattern is crucial for maintaining posture and stabilizing the spinal column [15]. During diaphragmatic breathing, the upper chest and accessory respiratory muscles remain relaxed. In contrast, thoracic or chest breathing is a dysfunctional pattern characterized by upper chest movement and additional muscle activation. Restricted thoracic spine mobility, rib immobility, or shortened respiratory muscles often cause this maladaptive pattern [16]. The breathing exercise protocols used in various studies emphasize optimal breathing mechanics, activation of the diaphragm, and engagement of key anatomical structures essential for improving mobility [17,18,19,20].

Techniques such as pranayama [21], pursed-lip breathing [22], and the Buteyko method [23], alongside mobility interventions like PNF stretching [24] and thoracic spine mobilization [25], have been applied to improve respiratory function and joint mobility. However, diaphragmatic breathing remains the established standard based on the principles of proper breathing mechanics, emphasizing expansion of the lower ribs, activation of the diaphragm, and minimal use of accessory muscles. All listed breathing techniques should follow proper mechanics to improve respiratory efficiency and postural control.

Breathing exercises promoting diaphragmatic breathing and proper mechanics can enhance diaphragm muscle fiber recruitment, improve mobility, facilitate lower chest expansion, and contribute to better lung function [2,26]. Regularly implementing these exercises to strengthen and increase the endurance of respiratory muscles may help establish an optimal breathing pattern, thereby improving lung function [10,27]. Additionally, improvements in vital capacity have been observed in individuals with scoliosis and specific respiratory dysfunctions [28]. Studies examining the effects of breathing exercises on posture, thoracic spine mobility, chest expansion, trunk muscle endurance, and lung function have reported positive outcomes across all these parameters [17,18]. In addition to these long-term benefits, acute improvements, such as enhanced shoulder girdle mobility and pain reduction, have also been documented following breathing exercise interventions [19].

Research on the effects of breathing exercises on the musculoskeletal, nervous, cardiovascular, and respiratory systems has gained increasing interest among researchers and sports professionals seeking to enhance health and athletic performance [29,30,31,32]. Most prior research has predominantly involved populations with pathological or postural conditions such as COPD, kyphosis, or forward head posture [33,34,35]. Although these studies highlight various benefits of breathing exercises, their focus on impaired populations limits preventive and performance-optimizing potential in individuals without impairments. In our study, the integration of pulmonary function measures with functional mobility assessments enabled a holistic evaluation of the impact of breathing exercises in healthy individuals. Moreover, despite increasing interest in breathing techniques within sports and clinical contexts, existing studies have predominantly focused either on their effects on athletic performance [30,31] or on managing clinical symptoms such as postural dysfunctions [34], with limited attention given to their direct impact on functional joint mobility, particularly in segments crucial for postural control and upper body function, such as the thoracic spine and shoulder girdle. Given their relevance in sports, rehabilitation, and daily life, this study aims to examine the acute effects of diaphragmatic breathing on chest, thoracic spine, and shoulder girdle mobility, as well as lung function, hypothesizing that positive effects will also be observed in a healthy population.

## 2. Materials and Methods

### 2.1. Design and Procedures

This experimental study employed an acute pretest–posttest design with random group assignment and homogenization. We performed the pre- and post-tests on the same day. We homogenized the groups based on chest expansion test results (subchapter—Mobility tests). We chose the chest expansion test for group homogenization due to its direct association with diaphragm function and breathing mechanics. The participants underwent anthropometric assessment (body height, body mass) at the beginning of the testing session, followed by mobility and pulmonary function tests. After the initial assessments, participants in the experimental group performed a 22 min diaphragmatic breathing intervention. In contrast, those in the control group stayed in the same position but did not consciously apply the intervention, which involved natural breathing. After the intervention, participants repeated the tests in the same order. We performed each test twice and used the best result for further analysis.

One kinesiology professional conducted the tests, and another led the intervention. Both gave standardized instructions and motivated participants equally. Every participant followed the experimental protocol independently of other participants, without knowing the study aim about the differences in effects between natural and diaphragmatic breathing. This randomized controlled trial study was conducted in accordance with the CONSORT guidelines. The study was reviewed and approved by the Ethics Committee of the Faculty of Sport and Physical Education, University of Belgrade (Approval Number: 02-207/25-2; Date: 6 March 2025). All participants provided written informed consent prior to participation. The study was conducted in accordance with the ethical principles of the Declaration of Helsinki.

### 2.2. Subjects

A priori sample size was calculated using G*Power software v3.1.9.7 (Heinrich Heine University, Düsseldorf, Germany). Analysis for a repeated-measures ANOVA with a between-subject factor (2 groups) and within-subject factor (2 measurements) indicated that a minimum of 16 participants (8 per group) would be required to achieve 80% statistical power and to detect large effect size f (0.4), at an alpha level of 0.05. The study sample consisted of 26 (13 per group) physically active male students from the Faculty of Sports and Physical Education (aged 24.3 ± 2.0 years, body height of 182.9 ± 6.4 cm, body weight of 82.8 ± 10.4 kg, and body mass index of 24.7 ± 2.5 kg/m^2^). Only male participants were included in the study to minimize variability due to anatomical and physiological differences between sexes, such as variations in lung volumes and capacities, thoracic cage dimensions, respiratory muscle strength, and differences in connective tissue elasticity affecting mobility [36,37,38]. The aim was to assess the effects of diaphragmatic breathing within a homogeneous group rather than to perform a sex-based comparison. Inclusion criteria required healthy and physically active male participants with no history of respiratory conditions, musculoskeletal injuries, or pain at the time of testing. Exclusion criteria included any current or past diagnosis of obstructive pulmonary diseases (such as asthma or COPD), neuromuscular disorders, ongoing pain conditions (back pain, joint pain), or recent injuries that could affect movement or breathing patterns.

### 2.3. Experimental Protocol

The protocol emphasized diaphragmatic breathing, explicitly targeting the activation of the diaphragm and three-dimensional chest expansion. The participants began in a bridge position with their legs in a 90/90° hip/knee configuration while holding a rubber ball between their knees. They maintained a supine position with their feet placed against a wall, ensuring neutral neck and spine alignment. This passive 90° hip and knee flexion position induced relative lumbar spine flexion, posterior pelvic tilt, and internal rotation and depression of the rib, which together facilitate optimal breathing mechanics and proper diaphragm activation. The inclusion of a rubber ball between the knees promoted adductor muscle engagement and co-contraction of the pelvic floor muscles [39].

The breathing protocol consisted of two 10 min sessions separated by a 2 min break to prevent potential dizziness or fainting (22 min in total). During the first session, participants placed one hand on the upper chest and the other on the abdominal wall, emphasizing movement in the lower chest and abdominal region (Figure 1A). They were instructed to focus on elevating the hand placed on the abdomen during inhalation, facilitating proper diaphragm engagement and maximizing lower chest expansion. During the second session, participants placed their hands on the sides of the lower chest (Figure 1B) to emphasize lateral chest expansion. In both sessions, during one respiratory cycle, participants inhaled for 3–4 s at 80–90% of their maximum capacity, followed by complete exhalation lasting 5–7 s and a brief pause of 2–3 s. This controlled breathing pattern promotes relaxation of the neuromuscular system and reduces overall muscle tone at rest [17]. The examiner supervised the entire breathing protocol to ensure proper execution.

### 2.4. Mobility Tests

#### 2.4.1. Chest Expansion Test

Previous research has demonstrated good intrarater and interrater reliability of the chest expansion test (CE) in healthy individuals [40]. The test was performed according to a standardized procedure [41], and the change in chest circumference between maximal inhalation and maximal exhalation was assessed. Measurements were taken at anatomical reference points—the processus xiphoideus and the processus spinosus of the tenth thoracic vertebra. The participants stood with their arms relaxed at their sides and were instructed to inhale slowly through the nose, expanding their lungs as much as possible. A complete exhalation followed this through the mouth. The chest circumference was recorded at the end of both maximal inhalation and maximal exhalation. The difference between the inspiratory and expiratory chest circumferences was calculated with a precision of 0.5 cm.

#### 2.4.2. Thoracic Spine Rotation Test

The thoracic spine rotation (TSR) test has demonstrated high reliability and low measurement error [42]. In this study, participants were seated with their hips and knees bent at 90° and their trunks positioned in an upright, neutral alignment. A sponge block was placed between the knees, and the participants were instructed to apply gentle pressure by compressing the block with the upper legs. This block was used to minimize lower-body rotation, which could interfere with thoracic spine rotation. The participant placed a round wooden stick over the shoulders and was instructed to perform maximal trunk rotation to one side. The angle formed between the horizontal line and the stick at the end of the range of motion was recorded. The results were obtained through photo analysis. The camera was positioned 2 m above the participant.

#### 2.4.3. Lateral Trunk Flexion Test

In the lateral trunk flexion test (LTF), participants stood with their backs against the wall, ensuring that their heels and buttocks were in contact with the wall. The shoulders were as low as possible to prevent shoulder elevation during movement. A wooden block was placed between the participants’ feet to provide a stable and supportive base. The amplitude was measured as participants performed maximal lateral flexion movement, pressing their palms against the upper or lower leg. The angle formed by the horizontal line and the line connecting the left and right acromion was recorded. The results were obtained through photo analysis. The camera was positioned at shoulder height, 3 m in front of the subject.

#### 2.4.4. Shoulder Girdle Mobility Test

During the shoulder girdle mobility (SGM) test, participants took an upright position with both hands clenched into fists and thumbs placed inside the fingers. The participants then simultaneously moved one hand behind the back and the other behind the neck, achieving maximum adduction, extension, and internal rotation in one shoulder. In contrast, the other shoulder achieved maximum abduction, flexion, and external rotation. The fists were moved simultaneously in a single motion without adjusting the hand position. The distance between the two closest points of the fists was measured to determine the participant’s reach. The results were obtained through photo analysis. The camera was positioned at a shoulder height of 3 m in front of the participant.

A Redmi Note 12 mobile phone with a 50 MP camera (8160 × 6144 resolution) was used for photography. Data were obtained via the valid and reliable movement analysis program Kinovea (v.0.9.5-x64, https://www.kinovea.org/) [43].

#### 2.4.5. Pulmonary Function Test

Lung function was assessed via a spirometer (Spirolab III, Medical International Research S.p.A., Rome, Italy). The test was performed while the participants were seated. Following the instructions, the participants inhaled as deeply as possible and then exhaled forcefully to empty their lungs. The participants leaned slightly forward during full exhalation to facilitate exhalation by compressing the abdomen. The detailed procedures are outlined in the Spirolab III user manual (Medical International Research). The following variables were obtained: forced vital capacity (FVC), forced expiratory volume in one second (FEV1), and the FEV1/FVC ratio.

### 2.5. Statistical Analysis

The normality of distribution of the tested variables within the groups was assessed via the Shapiro–Wilk test, whereas the homogeneity of variances between groups was evaluated via Levene’s test. The sphericity of the data was tested with Mauchly’s sphericity test, and when sphericity was violated, the Greenhouse–Geisser correction was applied. The mean value and standard deviation were calculated and are presented for each variable. A mixed between-within-subjects ANOVA was used to determine group differences and assess the effects of the applied intervention and the interaction of these factors. Post hoc tests were conducted via pairwise comparisons with Bonferroni correction. The alpha level of statistical significance was set to ≤0.05. Effect size (ES) was calculated via Cohen’s D as the difference between the pre-test and post-test mean values divided by the pooled standard deviation and as the difference between groups in post-test mean values divided by the pooled standard deviation. The effect size was considered small if it was ≤0.49, medium if it was between 0.50 and 0.79, and large if it was ≥0.80 [44]. Partial eta squared (pη^2^) for both factors and their interaction is also reported. Statistical analysis was performed with IBM SPSS Statistics 20 (IBM Corp., Armonk, New York, USA).

## 3. Results

Table 1 presents the results of the mixed between-within-subjects ANOVA, which examined the effects of group, time, and their interactions on mobility tests and pulmonary function. The Shapiro–Wilk test confirmed a normal data distribution (*p* > 0.05) for all variables. Homogeneity of variances was verified for all variables via Levene’s test (*p* > 0.05).

Mixed between-within-subjects ANOVA showed significant group-time interaction in CE, TSR, LTF, SGM, and FVC. No significant interaction was found in the FEV1 and FEV1/FVC tests; however, a trend toward statistical significance was observed in the FEV1/FVC test, along with significant effects of the group and time factors. In the FEV1 test, no statistical significance was found for any of the factors or their interactions. Descriptive statistics and post hoc analyses for the mobility and pulmonary function tests are presented in Figure 2, Figure 3, Figure 4, Figure 5, Figure 6, Figure 7 and Figure 8.

Group homogenization was successful: CE pre-test results showed no difference, and the mean values were similar between groups (Figure 2). Additionally, significant differences between the groups in the pre-test were not found for most variables, except for the TSR (Figure 3).

Post hoc analysis revealed a significant increase in the CE, TSR, LTF, SGM, and FVC (Figure 2, Figure 3, Figure 4, Figure 5 and Figure 6) and a significant decrease in the FEV1/FVC (Figure 8) in the experimental group following the diaphragmatic breathing exercise protocol. No significant change was observed in the FEV1 (Figure 7). No significant differences were observed between the pre-test and post-test scores in the control group for any variable. In the post-test, significant differences between the experimental and control groups were observed in the FEV1/FVC (Figure 8), and a trend was noted in the CE (Figure 2).

After the intervention in the experimental group, small effects were observed in the SGM_(L)_ and SGM_(R)_, FVC, and FEV1/FVC tests (ES = 0.38–0.47); moderate effects were observed in the CE, TSR_(L)_, and LTF_(L)_ tests (ES = 0.62–0.76); and large effects were observed in the TSR_(R)_ and LTF_(R)_ tests (ES = 0.84–1.00). Between the experimental and control groups in the post-test, a medium effect was observed in the CE test (ES = 0.75), and a large effect was detected in FEV1/FVC (ES = 1.19).

## 4. Discussion

This study examined the acute effects of diaphragmatic breathing on trunk and shoulder mobility and lung function. Consistent with the hypotheses and findings of previous research, the results indicate that a breathing intervention primarily targeting the activation of the lower chest and diaphragm can lead to an acute improvement in the mobility of the chest, thoracic spine, and shoulder girdle. Additionally, a significant improvement in forced vital capacity was observed following the intervention, whereas no changes in other indicators of lung function were recorded. These findings suggest that diaphragmatic breathing may benefit physically active populations without health issues.

In the CE test, a significant improvement in the experimental group’s results was observed (Figure 2), which is consistent with findings from previous studies examining these effects [17]. The difference between the pre-test and post-test scores was 22.2% in the experimental group. The post-test results also revealed a 22.2% difference between the experimental and control groups. In diaphragmatic breathing, the chest works as one unit, and expansion depends on rib and thoracic spine mobility. All the ribs are attached to the spinal column. However, only the upper seven are connected to the sternum via costal cartilage, allowing greater flexibility in the lower part of the chest. This enhanced flexibility enables better adaptation to the movements of the diaphragm, which likely accounts for the improvement in the mobility of this region following diaphragmatic breathing. During inhalation, the expansion of the ribcage facilitates the creation of more space for lung expansion. Additionally, diaphragmatic breathing reduces tension in accessory respiratory muscles, optimizing diaphragm function and consequently improving the efficiency of chest expansion during inspiration [45].

In the TSR test, a significant improvement was observed in the experimental group following the intervention (Figure 3), with a 21.7% increase on the left side and a 23.3% increase on the right side. Given the significantly higher pre-test values in the control group than in the experimental group, no significant differences were observed in the post-test results. Research investigating the effects of breathing exercises on posture and thoracic spine mobility supports these findings [17]. Optimal mobility of the thoracic spine plays a critical role in providing smooth force transmission between the upper body and the lumbar spine [46], providing trunk stability. The TSR test assesses mobility in this spine segment, with limited rotation potentially indicating chest rigidity that could negatively impact respiratory function and movement mechanics [47,48]. The observed improvement in thoracic spine mobility following diaphragmatic breathing may be attributed to enhanced chest expansion and increased mobility of the lower ribs, which lift and expand laterally during inhalation. This process likely reduces the rigidity of the thoracic segment [49]. Notably, the prolonged exhalation phase in the protocol may also help reduce muscle tension in this region [50]. During this phase, the parasympathetic nervous system is stimulated, leading to muscle relaxation through decreased excitability of alpha motor neurons. The reduction in thoracic muscle tension can be attributed to this effect [51].

The results of the LTF test revealed a significant improvement in the experimental group following the intervention (Figure 4), with an 11.7% increase on the left side and a 15.4% increase on the right side [17]. This study also confirmed that diaphragmatic breathing has an acute effect on improving SGM (Figure 5), with a 20.2% increase on the left side and a 21.5% increase on the right side, supporting the findings of previous studies [19]. Although there was no significant difference in the SGM between the experimental and control groups in the post-test, the percentage difference was 24.2% on the left and 33.5% on the right. The mobility of the shoulder girdle is critical for the proper functioning of the entire upper body. The functional mobility of the shoulder girdle is intrinsically linked to the mobility of the chest and thoracic spine, as the shoulders are anatomically connected to these structures. Thus, changes in posture and breathing mechanics due to limitations in chest mobility can negatively affect shoulder girdle function. These findings further highlight the importance of diaphragmatic breathing in restoring intermuscle coordination and enhancing functional movement in the thoracic and lumbar spine. In the context of breathing, the shoulder girdle enables proper lung expansion and function [19].

Shoulder girdle, chest, and thoracic spine anatomical connections explain how diaphragmatic breathing improves mobility. Therefore, incorporating exercise and techniques to enhance chest mobility can be an important component of a comprehensive approach to improving overall shoulder mobility and maintaining optimal upper body function. The observed improvement in mobility can also be attributed to the enhanced synergistic contraction of the diaphragm and its antagonist muscles, which optimizes thoraco-abdominal pressure and increases rib cage and spine flexibility and stability [52]. The improvements may also result from enhanced neural drive to the diaphragm motor units and coordinated recruitment of accessory respiratory muscles, optimizing chest wall expansion and enhancing proprioceptive feedback from the muscles and joints of the thorax [53].

Consistent with previous studies [26,27,54], it has been established that breathing exercises can improve lung function. However, these studies generally focused on long-term effects. Sivakumar et al. (2011) also demonstrated that specific lung function parameters could improve immediately following the application of such exercises [55]. The results of our study indicate a significant improvement in FVC of 4.7% in the experimental group (Figure 6, Figure 7 and Figure 8). In terms of the FEV1/FVC ratio, a significant decrease of 4.6% was noted between the pre-test and post-test values in the experimental group, whereas no such changes were observed in FEV1. Because FVC improved while FEV1 remained unchanged, their ratio decreased. Pulmonary function indicators reflect the airflow rate through the airways, which requires long-term adaptations [56]. This finding is consistent with previous research, where long-term application of breathing exercises led to changes in FEV1 and the FEV1/FVC ratio [26,27]. The observed improvements in lung function may be attributed to the combined contributions of several key factors. Prolonged exhalation, in particular, enhances the activation of the diaphragm, a critical muscle involved in forced exhalation [57]. This increased engagement of the diaphragm likely facilitated a more significant expansion of the chest wall, enhanced activation of the abdominal muscles, and more efficient expulsion of air from the lungs, leading to the observed increase in FVC.

The observed improvements in mobility and pulmonary function suggest that diaphragmatic breathing may be a practical and effective method for enhancing physical performance and respiratory efficiency in healthy young individuals. This breathing exercise intervention is simple, inexpensive, and requires no specialized equipment, making it applicable in many settings. Furthermore, it is suitable for individuals of all fitness levels, as it does not involve strenuous physical exertion, holding potential for broader application in both clinical and athletic populations. In rehabilitation settings, it may be beneficial for individuals with limited mobility or early signs of postural and respiratory dysfunction. On the other hand, integrating diaphragmatic breathing into warm-up or recovery routines in athletic training may support better core control, spinal mobility, and breathing economy, one of the key elements for performance enhancement and injury prevention.

A notable limitation of this study is the exclusive focus on the acute effects of the breathing exercises, without examining long-term adaptations. Additionally, the mobility tests involved assessment of specific regions using indirect testing methods. Finally, the exclusive use of smartphone-based image capture for photogrammetric analysis in Kinovea could have affected measurement accuracy due to potential inconsistencies in camera positioning and image quality. Future research should examine the long-term effects of different breathing exercises. Exploring various breathing protocols may provide deeper insight into the most effective methods for enhancing mobility and pulmonary function. Additionally, the neurophysiological and biomechanical mechanisms underlying the observed improvements could be valuable problems to be examined in further studies.

## 5. Conclusions

The results demonstrated that a 22 min diaphragmatic breathing intervention leads to significant acute improvements in the mobility of the chest, thoracic spine, and shoulder girdle, as well as in lung function, specifically forced vital capacity, in healthy young adults. These findings support the potential of diaphragmatic breathing as an effective, simple, and accessible technique to enhance physical performance and respiratory efficiency in this population.

## Figures and Tables

**Figure 1 jfmk-10-00325-f001:**
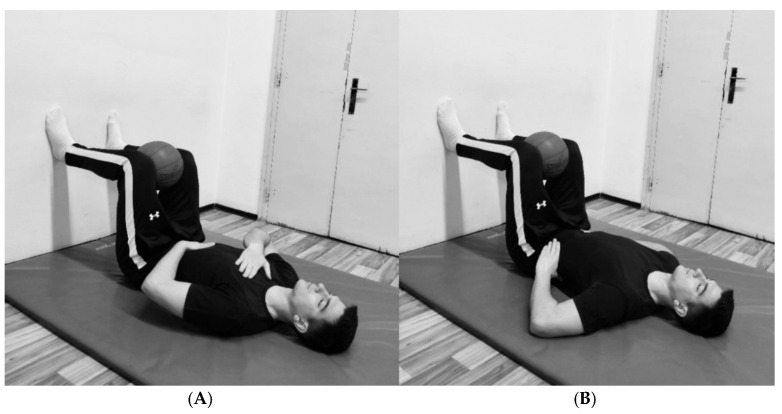
(**A**) Positions for the first session. (**B**) Position for the second session.

**Figure 2 jfmk-10-00325-f002:**
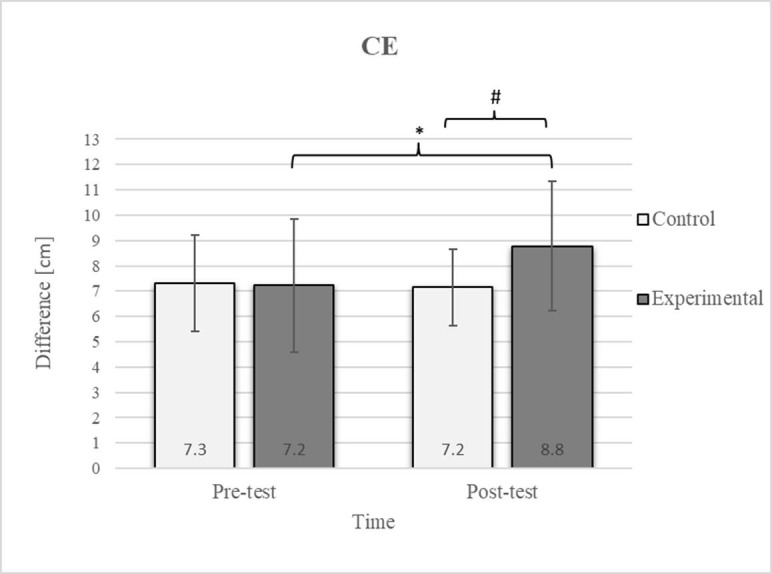
Descriptive statistics and post hoc analysis for chest expansion test at pre-test and post-test for control and experimental group (n = 13). *—statistically significant difference (*p* ≤ 0.05). #—tendency towards statistical significance (*p* ≈ 0.05).

**Figure 3 jfmk-10-00325-f003:**
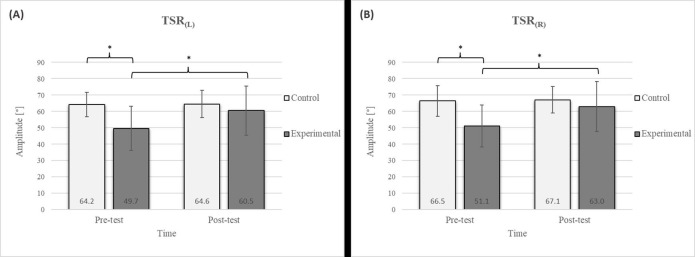
Descriptive statistics and post hoc analysis for thoracic spine rotation: (**A**) to the left and (**B**) to the right at pre-test and post-test for control and experimental group (n = 13). *—statistically significant difference (*p* ≤ 0.05).

**Figure 4 jfmk-10-00325-f004:**
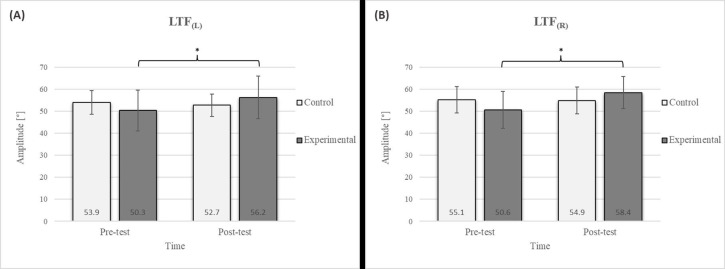
Descriptive statistics and post hoc analysis for lateral trunk flexion test: (**A**) to the left and (**B**) to the right at pre-test and post-test for control and experimental group (n = 13). *—statistically significant difference (*p* ≤ 0.05).

**Figure 5 jfmk-10-00325-f005:**
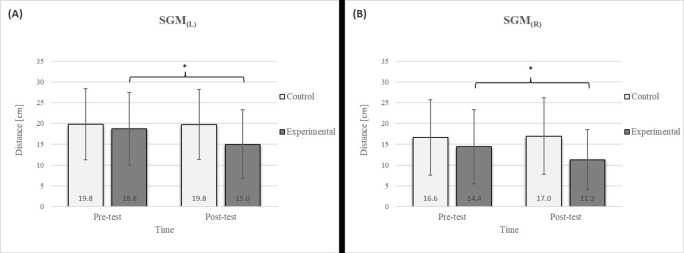
Descriptive statistics and post hoc analysis for shoulder girdle mobility test: (**A**) left side and (**B**) right side at pre-test and post-test for control and experimental group (n = 13). *—statistically significant difference (*p* ≤ 0.05).

**Figure 6 jfmk-10-00325-f006:**
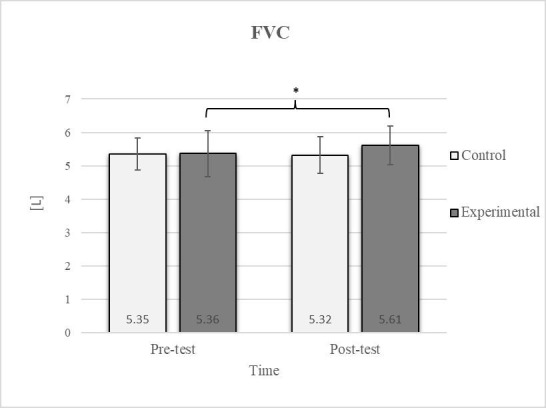
Descriptive statistics and post hoc analysis for force vital capacity at pre-test and post-test for control and experimental group (n = 13). *—statistically significant difference (*p* ≤ 0.05).

**Figure 7 jfmk-10-00325-f007:**
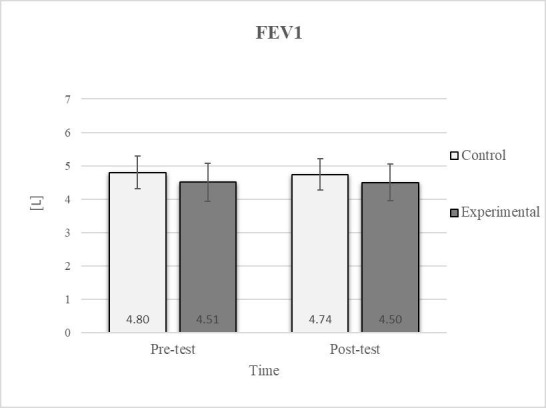
Descriptive statistics and post hoc analysis for force expiratory volume in first second at pre-test and post-test for control and experimental group (n = 13).

**Figure 8 jfmk-10-00325-f008:**
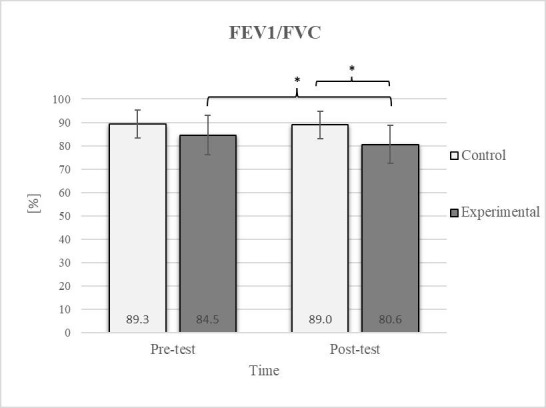
Descriptive statistics and post hoc analysis for ratio of force vital capacity and force expiratory volume in first second at pre-test and post-test for control and experimental group (n = 13). *—statistically significant difference (*p* ≤ 0.05).

**Table 1 jfmk-10-00325-t001:** Effect of the group factor (experimental vs. control), the time factor (pre-test vs. post-test), and their interaction on mobility tests and pulmonary function (n = 13).

	Group	Time	Group × Time
CE	F_(1)_ = 0.81 *p* = 0.38 *p*η^2^ = 0.03	F_(1)_ = 17.17 *p* ≤ 0.01 * *p*η^2^ = 0.42	F_(1)_ = 25.64 *p* ≤ 0.01 * *p*η^2^ = 0.52
TSR_(L)_	F_(1)_ = 4.22 *p* = 0.05 * *p*η^2^ = 0.15	F_(1)_= 79.72 *p* ≤ 0.01 * *p*η^2^ = 0.77	F_(1)_ = 70.35 *p* ≤ 0.01 * pη^2^ = 0.75
TSR_(R)_	F_(1)_ = 4.56 *p* ≤ 0.05 * *p*η^2^ = 0.16	F_(1)_ = 51.22 *p* ≤ 0.01 * *p*η^2^ = 0.68	F_(1)_ = 40.71 *p* ≤ 0.01 * *p*η^2^ = 0.63
LTF_(L)_	F_(1)_ = 0.00 *p* = 0.99 *p*η^2^ = 0.00	F_(1)_ = 7.31 *p* = 0.01 * *p*η^2^ = 0.23	F_(1)_ = 16.16 *p* ≤ 0.01 * *p*η^2^ = 0.40
LTF_(R)_	F_(1)_ = 0.04 *p* = 0.85 *p*η^2^ = 0.00	F_(1)_ = 21.74 *p* ≤ 0.01 * *p*η^2^ = 0.48	F_(1)_ = 25.13 *p* ≤ 0.01 * *p*η^2^ = 0.51
SGM_(L)_	F_(1)_ = 0.73 *p* = 0.40 pη^2^ = 0.03	F_(1)_ = 29.71 *p* ≤ 0.01 * *p*η^2^ = 0.55	F_(1)_ = 29.11 *p* ≤ 0.01 * pη^2^ = 0.55
SGM_(R)_	F_(1)_ = 1.31 *p* = 0.26 pη^2^ = 0.05	F_(1)_ = 12.26 *p* ≤ 0.01 * pη^2^ = 0.34	F_(1)_ = 18.77 *p* ≤ 0.01 * pη^2^ = 0.44
FVC	F_(1)_ = 0.46 *p* = 0.50 pη^2^ = 0.02	F_(1)_ = 6.41 *p* ≤ 0.05 * pη^2^ = 0.21	F_(1)_ = 9.52 *p* ≤ 0.01 * pη^2^ = 0.28
FEV1	F_(1)_ = 1.65 *p* = 0.21 *p*η^2^ = 0.06	F_(1)_ = 1.53 *p* = 0.23 *p*η^2^ = 0.06	F_(1)_ = 0.72 *p* = 0.41 *p*η^2^ = 0.03
FEV1/FVC	F_(1)_ = 5.96 *p* ≤ 0.05 * *p*η^2^ = 0.20	F_(1)_ = 5.45 *p* ≤ 0.05 * *p*η^2^ = 0.18	F_(1)_ = 3.87 *p* = 0.06 ^#^ *pη*2 = 0.14

*—statistically significant difference (*p* ≤ 0.05). #—tendency towards statistical significance (*p* ≈ 0.05). CE—chest expansion; FEV1—forced expiratory volume in first second; FVC—forced vital capacity; LTF—lateral trunk flexion; SGM—shoulder girdle mobility; TSR—thoracic spine rotation.

## Data Availability

The datasets used and/or analyzed during the current study are available from the corresponding author on reasonable request.

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
