# Peer review of "Acute Effects of Diaphragmatic Breathing on Trunk and Shoulder Mobility and Pulmonary Function in Healthy Young Adults"

_jfmk, 2025, doi:10.3390/jfmk10030325_

Round 1
Reviewer 1 Report
Comments and Suggestions for Authors
Thank you for the opportunity to review the paper "Acute effects of diaphragmatic breathing on trunk and shoulder mobility and pulmonary function in healthy young adults".
Relevant and interesting topic considering the population under study and respiratory function.
To make it easier for the authors to understand, I will review the paper section by section.
Abstract
In the methods in line 9, there seems to be a typographical error, where the same word ‘males’ is repeated twice.
Introduction
I enjoyed reading the introduction, however, I would like to see the relevance/justification for conducting this study clarified further. Could you improve this aspect?
Materials & Methods
I think it would be important to justify why only men were included in the study. Please review this information.
The inclusion and exclusion criteria are not clarified in the paper.
Regarding the homogeneity of the sample, what is the justification for only considering the chest expansion test? Wouldn't it be important to check other variables? It would be pertinent to reflect on this topic.
Which professionals gave instructions to the participants? It would be important to describe this in the paper.
Who administered the tests? One or more professionals? If more than one, did they verify intra-observer reliability?
Was a pilot study conducted? If so, this should be mentioned in the paper, along with its purpose.
Were the participants blind to the groups they were part of? And were the researchers also blind?
Results
Table 1 refers to which group is not clear.
In Table 1, care should be taken to use the same number of decimal places.
It would be important to include the value of p in Figures 2 to 11.
Couldn't they present the results in another way, without having 10 figures?
They should review the results, as there is a lot of information in the text that is repeated in the table or even in the graphs.
Discussion
The discussion is well organised, but it fails to address the possible limitations of the study. It would also be useful if the suggestions were more concrete and operational.
Conclusion
It should only respond to the objectives under study, as stated in the abstract.
Author Response
Comments 1: Thank you for the opportunity to review the paper "Acute effects of diaphragmatic breathing on trunk and shoulder mobility and pulmonary function in healthy young adults".
Relevant and interesting topic considering the population under study and respiratory function.
To make it easier for the authors to understand, I will review the paper section by section.
Response 1: Dear reviewer, thank you for your valuable time in considering this paper, as well as your helpful comments that improved the paper. In the following text we addressed your every comment and responded to it with confirmation and/or proper justification. Furthermore, for your convenience our responses are bolded and every modified part in the manuscript is marked in red. We are very grateful for your honest support of publishing our work!
Comments 2: In the methods in line 9, there seems to be a typographical error, where the same word ‘males’ is repeated twice.
Response 2: Corrected in the text (line 9).
Comments 3: I enjoyed reading the introduction, however, I would like to see the relevance/justification for conducting this study clarified further. Could you improve this aspect?
Response 3: We further elaborated the approach to this research problem and properly referenced it. It can be found at the end of the introduction section (line 92-104).
Comments 4: I think it would be important to justify why only men were included in the study. Please review this information.
Response 4: We now addressed the lack of this explanation by stating the anatomical and physiological differences between males and females related to breathing process. It can be found in the Subjects subsection (line 142-147).
Comments 5: The inclusion and exclusion criteria are not clarified in the paper.
Response 5: We now added these criteria in the Subjects section (line 147-152).
Comments 6: Regarding the homogeneity of the sample, what is the justification for only considering the chest expansion test? Wouldn't it be important to check other variables? It would be pertinent to reflect on this topic.
Response 6: We now provided the explanation for considering only the chest expansion test as the key variable for homogenization criteria. This addition can be found at the beginning of the Design and procedures subsection (line 114-116). We believe that homogenization is sufficient on the main variable as long as the other variables are not significantly different in the pre-test.
Comments 7: Which professionals gave instructions to the participants? It would be important to describe this in the paper.
Response 7: In the second paragraph of Design and procedures subsection we clarified the personnel being in charge of the intervention and testing (line 124-125).
Comments 8: Who administered the tests? One or more professionals? If more than one, did they verify intra-observer reliability?
Response 8: The response to previous question also covers these questions (line 124-128).
Comments 9: Was a pilot study conducted? If so, this should be mentioned in the paper, along with its purpose.
Response 9: Pilot study was not conducted, but before conducting this study we tested on 3 subjects (that are not included in the sample) the whole study protocol. Since the protocol itself wasn’t very hard to conduct, we found it acceptable to do the whole study at once without the pilot study.
Comments 10: Were the participants blind to the groups they were part of? And were the researchers also blind?
Response 10: Also addressed in the second paragraph of the Design and procedures subsection (line 125-128).
Comments 11: Table 1 refers to which group is not clear.
Response 11: We are sorry if we made it sound confounding. We modified the name of the Table 1 so it would be clear to the reader that the group refers to the factor (line 256-257).
Comments 12: In Table 1, care should be taken to use the same number of decimal places.
Response 12: Corrected in the Table 1, as well as for p values that now show in two decimals the significance.
Comments 13: It would be important to include the value of p in Figures 2 to 11.
Response 13: In every figure legend we added the value of p bellow which the results is considered significant (line 270, 271, 275, 283, 287, 290, 291, 298, 302, 303).
Comments 14: Couldn't they present the results in another way, without having 10 figures?
Response 14: The best we could do was to merge two figures in one figure and separate it with (A) and (B) wherever the left and the right side results are provided (line 272, 280, 284). In this case, we now have 7 figures (line 267) instead of 10 and we modified the figure numbering throughout the results and discussion sections as well. We truly hope this is satisfactory now.
Comments 15: They should review the results, as there is a lot of information in the text that is repeated in the table or even in the graphs.
Response 15: We assume this comment is related to the p values being repeated in the text after the figures and we deleted it in the text (line 304-314). On the other hand, we kept the original description of results since it summarized and grouped the individiual results according to the level of significance and effect size.
Comments 16: The discussion is well organised, but it fails to address the possible limitations of the study. It would also be useful if the suggestions were more concrete and operational.
Response 16: Limitations are further expanded, as well as suggestions and practical applications in different settings (line 404-421).
Comments 17: It should only respond to the objectives under study, as stated in the abstract.
Response 17: We now concluded in accordance to the objectives. It is now more straightforward conclusion thanks to this suggestion (line 428-433).

Reviewer 2 Report
Comments and Suggestions for Authors
The manuscript explores a relevant and timely topic, investigating the acute effects of diaphragmatic breathing on thoracic and shoulder mobility, as well as pulmonary function, in healthy adults. The subject matter is well aligned with current interests in rehabilitation, sports science, and respiratory health. However, to meet the level of scientific rigor expected for a journal with an impact factor of 3, the manuscript would benefit from several substantial revisions. Below, I outline key areas that require attention:
While the background provides useful context, the manuscript would benefit from a clearer articulation of the specific scientific gap being addressed. The rationale for selecting this particular breathing protocol, as well as the choice of a healthy male-only sample, needs to be better justified. Additionally, comparing this approach with alternative breathing or mobility interventions could help position the study more effectively within the existing literature.
The study is designed as a randomized trial with a control group, which is appropriate. However, the relatively small sample size (n = 13 per group) and the inclusion of only male participants, without a stated justification for excluding women, limits the generalizability of the findings. Furthermore, the absence of blinding during outcome assessments may have introduced bias, particularly given the subjective nature of some mobility measurements.
The methods are generally well described. Still, a key limitation is the absence of objective confirmation that the diaphragmatic breathing protocol effectively activated the diaphragm (e.g., using electromyography or ultrasound). This omission raises questions about the internal validity of the intervention. Additionally, while the use of Kinovea for photogrammetric analysis is appropriate, the limitations of relying solely on smartphone-based image capture were not sufficiently acknowledged or discussed.
The results are clearly structured, but there is some redundancy in how data are presented — several figures appear to duplicate information already included in tables. Moreover, the manuscript does not explore the clinical or functional significance of the observed effect sizes. Including a post-hoc power analysis and contextualizing smaller effects in terms of real-world impact would strengthen this section.
The discussion appropriately emphasizes anatomical mechanisms, but it would benefit from deeper engagement with neurophysiological and biomechanical explanations for the observed improvements. A more balanced interpretation—including a reflection on potential confounding factors and study limitations—is also needed. Finally, expanding on the practical implications of the findings for clinical or athletic populations would add value.
The manuscript would benefit from language editing. Several sentences are overly long, and there is frequent use of passive voice, which affects clarity and flow. A professional review by a native English speaker or editing service is strongly recommended to enhance readability and precision.
Overall, the study presents interesting and potentially impactful findings. However, revisions are necessary—particularly in the framing of the introduction, methodological transparency, discussion depth, and language style—to elevate the manuscript to the standards expected for publication.
Author Response
Comments 1: The manuscript explores a relevant and timely topic, investigating the acute effects of diaphragmatic breathing on thoracic and shoulder mobility, as well as pulmonary function, in healthy adults. The subject matter is well aligned with current interests in rehabilitation, sports science, and respiratory health. However, to meet the level of scientific rigor expected for a journal with an impact factor of 3, the manuscript would benefit from several substantial revisions. Below, I outline key areas that require attention:
Response 1: Dear reviewer, thank you for your valuable time in considering this paper, as well as your helpful comments that improved the paper. In the following text we addressed your every comment and responded to it with confirmation and/or proper justification. Furthermore, for your convenience our responses are bolded and every modified part in the manuscript is marked in red.
Comments 2: While the background provides useful context, the manuscript would benefit from a clearer articulation of the specific scientific gap being addressed. The rationale for selecting this particular breathing protocol, as well as the choice of a healthy male-only sample, needs to be better justified. Additionally, comparing this approach with alternative breathing or mobility interventions could help position the study more effectively within the existing literature.
Response 2: We further elaborated the approach to this research problem and properly referenced it. It can be found at the end of the introduction section (line 92-104). Regarding the chosen breathing protocol, we added one paragraph in the middle of the introduction that positiones the diaphgragmatic breathing in relation to the other protocols applied in other research settings (line 71-77). Moreover, we addressed the lack of explanation for including only male participants by stating the anatomical and physiological differences between males and females related to breathing process. It can be found in the Subjects subsection (line 142-147). The choice of including only healthy subjects is stated in the Introduction as being one of the study aims (line 104-108).
Comments 3: The study is designed as a randomized trial with a control group, which is appropriate. However, the relatively small sample size (n = 13 per group) and the inclusion of only male participants, without a stated justification for excluding women, limits the generalizability of the findings. Furthermore, the absence of blinding during outcome assessments may have introduced bias, particularly given the subjective nature of some mobility measurements.
Response 3: In the Subjects subsection we showed the results of G*Power analysis that indicated that the sample size was sufficient for this analysis (line 135-139). In the Design and procedures subsection we clarified the personnel being in charge of the intervention and testing, as well as blinding issue related to participants (line 124-128). The explanation for including only male participants is covered in the response to the previous comment (line 142-147).
Comments 4: The methods are generally well described. Still, a key limitation is the absence of objective confirmation that the diaphragmatic breathing protocol effectively activated the diaphragm (e.g., using electromyography or ultrasound). This omission raises questions about the internal validity of the intervention. Additionally, while the use of Kinovea for photogrammetric analysis is appropriate, the limitations of relying solely on smartphone-based image capture were not sufficiently acknowledged or discussed.
Response 4: At the beginning of the 3rd paragraph in Introduction we clarified the diaphragm and other respiratory muscles involvement in the diaphragmatic breathing by referencing the direct methods of muscle assessment (line 59-62). Regarding the usage of smartphone-based image capture, we now pointed out this issue as a limitation of the study (line 418-421).
Comments 5: The results are clearly structured, but there is some redundancy in how data are presented — several figures appear to duplicate information already included in tables. Moreover, the manuscript does not explore the clinical or functional significance of the observed effect sizes. Including a post-hoc power analysis and contextualizing smaller effects in terms of real-world impact would strengthen this section.
Response 5: We are sorry if we made it sound confounding. We checked again and the only table in the manuscript is the Table 1 which refers to the results of ANOVA analysis, while figures present the descriptive statistics and post-hoc analysis. With regards to the clinical and functional significance, we are aware of its importance, but since we conducted a study with only acute effects, we find it difficult to include additional aspects important and sensitive to the long-term application. Additionally, we used the sample of healthy individuals that do not have any important deficits and insufficiencies.
Comments 6: The discussion appropriately emphasizes anatomical mechanisms, but it would benefit from deeper engagement with neurophysiological and biomechanical explanations for the observed improvements. A more balanced interpretation—including a reflection on potential confounding factors and study limitations—is also needed. Finally, expanding on the practical implications of the findings for clinical or athletic populations would add value.
Response 6: In the 5th paragraph of the Discussion section, we breafly introduced alternative explanations for the given study results (line 380-386). We found it difficult to expand more on it much since we haven’t dealt with neurophysiological aspects and the explanations could only be referenced according to other sources. Limitations are further expanded, as well as suggestions and practical applications in different settings (404-421).
Comments 7: The manuscript would benefit from language editing. Several sentences are overly long, and there is frequent use of passive voice, which affects clarity and flow. A professional review by a native English speaker or editing service is strongly recommended to enhance readability and precision.
Response 7: We further improved the manuscript text and all the changes are in the following lines: 10, 12-15, 23-24, 40-41, 121-123, 160, 261-262, 304-305, 334-335, 376-377, 421-422.
Comments 8: Overall, the study presents interesting and potentially impactful findings. However, revisions are necessary—particularly in the framing of the introduction, methodological transparency, discussion depth, and language style—to elevate the manuscript to the standards expected for publication.
Response 8: We are very grateful for your honest support of publishing our work!

Reviewer 3 Report
Comments and Suggestions for Authors
The subject of the article is interesting for readers and for the literature of the field even if it is not new.
The article has a good structure. It presents the introduction, method, results, discussions and references. The method part is well presented. The article has a scientific sound and is accompanied by statistical analysis.
I think that to improve the article, some explanations should be added regarding:
- how was the number of subjects established at 26 and not higher or lower?
- why were anthropometric measurements performed?
- why were measurements regarding body composition performed?
- do these measurements influence the respiratory protocol?
- in the Design and procedures part it is specified that "experimental group performed a 20-minute diaphragmatic breathing protocol" but in the Experimental protocol part it is specified that "the breathing protocol consisted of two 10-minute sessions separated by a 2-minute break" which means, in total, 22 minutes as a procedure (the break being part of the procedure). I believe that a revision is needed to present a unified opinion.
The article can be considered for publication after these minor changes have been clarified.
Author Response
Comments 1: The subject of the article is interesting for readers and for the literature of the field even if it is not new.
The article has a good structure. It presents the introduction, method, results, discussions and references. The method part is well presented. The article has a scientific sound and is accompanied by statistical analysis.
I think that to improve the article, some explanations should be added regarding:
Response 1: Dear reviewer, thank you for your valuable time in considering this paper, as well as your helpful comments that improved the paper. In the following text we addressed your every comment and responded to it with confirmation and/or proper justification. Furthermore, for your convenience our responses are bolded and every modified part in the manuscript is marked in red.
Comments 2: - how was the number of subjects established at 26 and not higher or lower?
Response 2: In the Subjects subsection we showed the results of G*Power analysis that indicated that the sample size was sufficient for this analysis (line 135-139).
Comments 3: - why were anthropometric measurements performed?
Response 3: We are sorry for our previous unclear description. In the brackets we now stated that we only measured body height and body mass in the anthropometric space (line 117).
Comments 4: - why were measurements regarding body composition performed?
Response 4: Since we didn’t report these variables, we now excluded this step in the study procedure.
Comments 5: - do these measurements influence the respiratory protocol?
Response 5: If your question is related to the previous questions, we believe that anthropometric and body composition measurements did not influence the intervention protocol since it is independent. The protocol is explained in detail.
Comments 6: - in the Design and procedures part it is specified that "experimental group performed a 20-minute diaphragmatic breathing protocol" but in the Experimental protocol part it is specified that "the breathing protocol consisted of two 10-minute sessions separated by a 2-minute break" which means, in total, 22 minutes as a procedure (the break being part of the procedure). I believe that a revision is needed to present a unified opinion.
Response 6: We corrected every place where 20-minutes intervention is stated to 22 minutes with regards to the pause being a part of the intervention as you correctly stated (line 11, 22, 119, 164, 428).
Comments 7: The article can be considered for publication after these minor changes have been clarified.
Response 7: We are very grateful for your honest support of publishing our work!

Reviewer 4 Report
Comments and Suggestions for Authors
Thank you for the invitation to review the article “Acute effects of diaphragmatic breathing on trunk and shoulder mobility and pulmonary function in healthy young adults”. The presented work is interesting, and the topic may be important for biomedical professionals. In general, this is a well-written study with good design and appropriate reporting. However, some revisions seem warranted. Importantly, I believe all of them are doable, as they are not associated with the rationale, scientific basis, or execution of the study.
MAJOR
The CONSORT guidelines for reporting RCTs should be applied and referenced. These are standardized aspects of reporting studies to improve transparency and reproducibility, and they should be applied to ensure a complete and unbiased presentation of research findings. Although this is only one point in my review, it concerns multiple aspects of the paper.
The rationale for introducing diaphragmatic breathing should be expanded with psychophysiological aspexcts, for example underline the role of hyperpnoea with diapgragmatic emphasis on lowering cortisol: https://pubmed.ncbi.nlm.nih.gov/39299616/ or effectiveness of diaphragmatic breathing for reducing physiological and psychological stress (DOI: 10.11124/JBISRIR-2017-003848).
You must use ANOVA for stats, not t-tests. Please adhere to: https://www.statology.org/how-to-report-two-way-anova-results/ when introducing and reporting statistical methods and results across the whole paper. With such a group partial eta squared and omega squared effect sizes must be reported.
MINOR:
- describe, briefly, the anatomy and function of respiratory muscle with underlining role of the diaphragm in the introduction
- since tables and figures must be self explanatory add n= for each group there in the headlines,
- line 199 - typo in post-test
- too many figures, combine them into less figures with panels a,b,c, etc.
- limitations of the presented study should be expanded, especially in context of cross-sectional study design
- think about reversing sections 2.1 and 2.2, soft suggestion
Author Response
Comments 1: Thank you for the invitation to review the article “Acute effects of diaphragmatic breathing on trunk and shoulder mobility and pulmonary function in healthy young adults”. The presented work is interesting, and the topic may be important for biomedical professionals. In general, this is a well-written study with good design and appropriate reporting. However, some revisions seem warranted. Importantly, I believe all of them are doable, as they are not associated with the rationale, scientific basis, or execution of the study.
Response 1: Dear reviewer, thank you for your valuable time in considering this paper, as well as your helpful comments that improved the paper. In the following text we addressed your every comment and responded to it with confirmation and/or proper justification. Furthermore, for your convenience our responses are bolded and every modified part in the manuscript is marked in red. We are very grateful for your honest support of publishing our work!
Comments 2: The CONSORT guidelines for reporting RCTs should be applied and referenced. These are standardized aspects of reporting studies to improve transparency and reproducibility, and they should be applied to ensure a complete and unbiased presentation of research findings. Although this is only one point in my review, it concerns multiple aspects of the paper.
Response 2: For this type of study we went through the checklist and stated in the Design and procedures subsection that is conducted in accordance with the CONSORT guidelines (line 128-129).
Comments 3: The rationale for introducing diaphragmatic breathing should be expanded with psychophysiological aspexcts, for example underline the role of hyperpnoea with diapgragmatic emphasis on lowering cortisol: https://pubmed.ncbi.nlm.nih.gov/39299616/ or effectiveness of diaphragmatic breathing for reducing physiological and psychological stress (DOI: 10.11124/JBISRIR-2017-003848).
Response 3: In the beginning of the Introduction we added the text according to this suggestion (line 32-34).
Comments 4: You must use ANOVA for stats, not t-tests. Please adhere to: https://www.statology.org/how-to-report-two-way-anova-results/ when introducing and reporting statistical methods and results across the whole paper. With such a group partial eta squared and omega squared effect sizes must be reported.
Response 4: We are sorry if we made it confounding. As already stated in the Statistical analysis subsection, we did use mixed between-within-subjects ANOVA (not t-tests) (line 240). We further clarified that we used “pairwise comparisons with Bonferroni correction” as a post-hoc (line 242-243), since “the pairwise comparisons” were previously not mentioned. Additionally, in Table 1 we added partial eta squared as effect size.
Comments 5: describe, briefly, the anatomy and function of respiratory muscle with underlining role of the diaphragm in the introduction
Response 5: In the 2nd paragraph of Introduction we expanded on the function of respiratory muscles that are involved in inspiration, passive and forced exhalation (line 46-50, 55-58).
Comments 6: since tables and figures must be self explanatory add n= for each group there in the headlines,
Response 6: Added in the description of Table 1 and every figure.
Comments 7: line 199 - typo in post-test
Response 7: Corrected in the text (now is line 244-246, on two separate places).
Comments 8: too many figures, combine them into less figures with panels a,b,c, etc.
Response 8: The best we could do was to merge two figures in one and separate it with (A) and (B) wherever the left and the right side results are provided (line 272, 280, 284). In this case, we now have 7 figures (line 267) instead of 10 and we modified the figure numbering throughout the results and discussion sections as well. We truly hope this is satisfactory now.
Comments 9: limitations of the presented study should be expanded, especially in context of cross-sectional study design
Response 9: Limitations are now further expanded (line 416-421). In the Design and procedures subsection we added a statement that it is acute study (along with the statement „pretest‒posttest design“) (line 112), to mitigate the potential misunderstandings in regards to the exact study design.
Comments 10: think about reversing sections 2.1 and 2.2, soft suggestion
Response 10: Reversed in the text (line 111 and 134).

Round 2
Reviewer 4 Report
Comments and Suggestions for Authors
I am happy to accept the paper, thank you for your work.
I have one more formatting ecommendation that hopefully might be introduced during the proofreading stage. You do use partial eta squared according to the Methods section but you use Eta2 abbreviations in the tables. The common way to present it is to add p befor, so pEta2 seems warranted in the results section.